# *EHBP1L1* Frameshift Deletion in English Springer Spaniel Dogs with Dyserythropoietic Anemia and Myopathy Syndrome (DAMS) or Neonatal Losses

**DOI:** 10.3390/genes13091533

**Published:** 2022-08-26

**Authors:** Sarah Østergård Jensen, Matthias Christen, Veronica Rondahl, Christopher T. Holland, Vidhya Jagannathan, Tosso Leeb, Urs Giger

**Affiliations:** 1Clinical Pathology Laboratory, The Swedish University of Agricultural Sciences, 750 07 Uppsala, Sweden; 2AniCura Small Animal Referral Hospital Bagarmossen, Ljusnevägen 17, Bagarmossen, 128 48 Stockholm, Sweden; 3Institute of Genetics, Vetsuisse Faculty, University of Bern, Bremgartenstrasse 109a, 3001 Bern, Switzerland; 4BioVet AB, 192 21 Sollentuna, Sweden; 5Merewether Veterinary Hospital, Suite 2, 25 Llewellyn St, Merewether, NSW 2291, Australia; 6Vetsuisse Faculty, University of Zürich, Winterthurerstrasse 260, 8057 Zürich, Switzerland

**Keywords:** *Canis lupus familiaris*, canine, metarubricytosis, microcytosis, centronuclear myopathy, megaesophagus, cardiomyopathy, perinatal death, precision medicine, animal model

## Abstract

Hereditary myopathies are well documented in dogs, whereas hereditary dyserythropoietic anemias are rarely seen. The aim of this study was to further characterize the clinical and clinicopathological features of and to identify the causative genetic variant for a dyserythropoietic anemia and myopathy syndrome (DAMS) in English springer spaniel dogs (ESSPs). Twenty-six ESSPs, including five dogs with DAMS and two puppies that died perinatally, were studied. Progressive weakness, muscle atrophy—particularly of the temporal and pelvic muscles—trismus, dysphagia, and regurgitation due to megaesophagus were observed at all ages. Affected dogs had a non-regenerative, microcytic hypochromic anemia with metarubricytosis, target cells, and acanthocytes. Marked erythroid hyperplasia and dyserythropoiesis with non-orderly maturation of erythrocytes and inappropriate microcytic metarubricytosis were present. Muscle biopsies showed centralized nuclei, central pallor, lipocyte infiltrates, and fibrosis, which was consistent with centronuclear myopathy. The genome sequencing of two affected dogs was compared to 782 genomes of different canine breeds. A homozygous frameshift single-base deletion in *EHBP1L1* was identified; this gene was not previously associated with DAMS. Pedigree analysis confirmed that the affected ESSPs were related. Variant genotyping showed appropriate complete segregation in the family, which was consistent with an autosomal recessive mode of inheritance. This study expands the known genotype–phenotype correlation of *EHBP1L1* and the list of potential causative genes in dyserythropoietic anemias and myopathies in humans. *EHBP1L1* deficiency was previously reported as perinatally lethal in humans and knockout mice. Our findings enable the genetic testing of ESSP dogs for early diagnosis and disease prevention through targeted breeding strategies.

## 1. Introduction

Pathogenic variants in a distinct gene frequently cause a particular metabolic pathway to fail and lead to specific clinical signs and organ dysfunction with the possibility of secondary effects. However, some gene defects can have pleiotropic effects and affect multiple organ systems, leading to a variety of clinical signs, which are often referred to as syndromes. Pleiotropy has been described in many hereditary diseases of humans and dogs [1,2,3]. Genetic pleiotropism may form unique syndromes, which are frequently named after the discoverer, the first patient, or the clinicopathologic manifestations. Some of the clinical signs may be seen at birth (congenital, including stillborn and neonatal deaths), while others develop later in life, depending on the specific gene involved, organ systems affected, and degree of dysfunction. Only rarely are perinatal deaths and progressive disease seen in a species with the same gene variant. With the advent of more detailed genetic analyses and the availability of better expression and functional studies, more pleiotropic gene effects have been documented and more light has been shed on the complex *in vivo* functions of many genes.

In dogs, several hundred hereditary disorders have been described, from clinical signs to gene defects [4,5,6]. For example, there are several well-characterized erythrocytic enzyme and membrane defects, but as yet, there are no reports of hemoglobinopathies [7,8,9]. Several hereditary dystrophic and other myopathies have been characterized in dogs [10,11,12,13,14]. However, it is rare that both erythrocytes and myocytes are affected by the same pleiotropic gene defect. A classic example is M-type phosphofructokinase deficiency (also known as glycogenosis type VII or Tarui–Layzer syndrome in humans), resulting in severe hemolytic crises and exertional metabolic myopathy in English springer spaniel (ESSP) dogs and a few other canine breeds [7,15,16].

Dyserythropoiesis and polymyopathy were reported in three ESSPs in Australia more than three decades ago [17]. Recently, a similar syndrome was also described in two Labrador retrievers in the United States [18]. We found similarly affected ESSPs in a large family in Sweden, which prompted us to further characterize this dyserythropoietic anemia and myopathy syndrome (DAMS), investigate neonatal deaths, and search for a likely single causative genetic defect.

## 2. Animals, Materials, and Methods

### 2.1. Animal Selection

The ESSP dog breed is divided into “show/bench” and “field/working” lines [19]. After the discovery of the index case, a four-year-old, neutered female show ESSP (Dog #1) with hematological and myopathic manifestations, one veterinarian (S.O.J.) examined all affected individuals and the extended family of show ESSPs at the AniCura Small Animal Referral Hospital Bagarmossen, Stockholm, Sweden. A Swedish registration number and microchip code was available for each examined dog, and a scientific pedigree was generated and searched for common ancestry within the databases of the Swedish Kennel Club [20] and the private spaniel database Fleckenbase [21]. In addition, pedigree information of the three (dogs #A1–A3) similarly affected previously reported ESSPs from Australia [17] and formalin-fixed, paraffin-embedded (FFPE) tissue blocks from two of those cases were available.

The total number of ESSPs included in the study was 45, of which 26 formed the extended family of Dog #1. The family of 26 dogs included five affected with DAMS, two that were perinatally deceased, and 19 unaffected show ESSPs from Sweden. Furthermore, two of the previously studied dogs from Australia affected by DAMS, which were distantly related to the Swedish dogs, were also included (Figure 1) [17]. Four clinically unaffected ESSPs that were not related to the affected dogs (>six generations), were seen for other reasons at the clinic in Sweden and were included in the hematological and biochemical analyses as breed controls, but not in the genetic analysis. The remaining DNA samples, which were derived from 13 ESSPs from Switzerland and Germany, served as breed controls, but without known relations to the extended-pedigree ESSPs with DAMS and without clinical information.

### 2.2. Samples and Data Collection

Information on ESSPs in Sweden was compiled from owners and breeders, with medical records that included signalment, medical history, clinical examination, radiographic results, and laboratory test results. A maximal volume of 10 mL of blood from affected, related, and unrelated ESSPs was collected in ethylenediaminetetraacetic acid (EDTA) for complete blood cell count (CBC), microscopic examination, and molecular genetic analyses, and it was collected in serum tubes for all other analyses. Blood smears were immediately prepared from EDTA blood. Leftover serum and EDTA blood samples were kept frozen for possible further analyses. The clinicopathological data for the three affected ESSPs from Australia were previously reported [17], and thus, they were not included in this study.

A clinical diagnosis of DAMS was made based upon the presence of (1) a non-regenerative, microcytic hypochromic anemia with marked metarubricytosis, which was determined with automated hematology analyses and microscopic blood smear evaluation, and (2) progressive skeletal muscle atrophy (particularly of the temporal and limb muscles) that was histopathologically supportive of centronuclear myopathy (CNM), when available.

### 2.3. Laboratory Diagnostics

An automated hematology analyzer (Sysmex xt2000i, Sysmex Nordic, Landskrona, Sweden), a microscopic evaluation of blood smears stained with May–Grünwald–Giemsa, and a Cobas C311 serum chemistry analyzer (Roche Diagnostics Scandinavia, Bromma, Sweden) were used. Urinalysis and the serum total thyroxin, cortisol, acetylcholine receptor antibodies, anti-nuclear antibody, lead, iron saturation, ferritin, soluble transferrin receptor, hepcidin (special serum iron parameters were determined at Laboklin, Bad Kissingen, Germany), C-reactive protein, trypsin-like immunoreactivity, and cobalamin parameters were also determined through routine veterinary diagnostic testing for a few affected and control ESSPs.

### 2.4. Gross Pathology and Histopathology

All five ESSPs with DAMS (Dogs #1–5) were euthanized because of progressive illness, but only two affected dogs (Dogs #2 and 4) were necropsied. The dogs were five and one years of age, respectively, and the necropsies were performed at different laboratories (National Veterinary Institute and University of Agricultural Sciences, both in Uppsala, Sweden). In addition, skeletal muscle biopsies were available from the index case, which was sampled at four years of age. Transversal and longitudinal muscle sections were prepared and processed according to standard protocols (4–5 µm; Meyer’s hematoxylin and eosin [H&E] sections and some special stains).

A software program was applied for microscopic image collection and quantification of 100 myofiber morphologies (IC 2.0.0.286) at 200× magnification in both longitudinal and transverse sections. The percentages of myofibers with centralized nuclei and myofibers with rows of internalized nuclei were calculated in longitudinal and transverse preparations, respectively.

### 2.5. Statistical Analysis

Descriptive statistics of clinicopathological analyses were calculated for continuous variables, and the results were tested for normality using the Shapiro–Wilk test. Box-plot figures were generated for the blood test parameters (RStudio, PBC, Boston, MA, USA), and a Kruskal–Wallis test was used to compare clinicopathological data from affected dogs to the data from healthy related and unrelated (control) ESSP dogs (Analyse-it^®^ for Microsoft Excel Office 365 software, Leeds, UK). A p-value of less than 0.05 was considered significant.

### 2.6. DNA Extraction and SNV Genotyping

Genomic DNA was extracted from EDTA blood and tissue samples according to the respective standard methods for EDTA blood, native tissue, and FFPE tissue using a Maxwell RSC instrument (Promega, Dübendorf, Switzerland). DNA from five affected and 19 unaffected animals was genotyped on Illumina_HD canine BeadChips containing 220,853 markers (Neogen, Lincoln, NE, USA). The raw SNV genotypes are available in Appendix A. The dogs that were genotyped with this method are indicated in Appendix A.

### 2.7. Linkage Analysis and Homozygosity Mapping

For the linkage analysis, a single ESSP family consisting of both parents and six puppies, including Dogs #4 and 5, was used. For all dogs, the call rate was >95%. Using PLINK v1.9 [22], markers that were non-informative, located on the sex chromosomes, missing in any of the dogs, and that had Mendel errors or a minor allele frequency of <0.01 were removed. The final pruned dataset contained 82,251 markers. To analyze the data for parametric linkage, an autosomal recessive inheritance model with full penetrance, a disease allele frequency of 0.3, and the Merlin software [23] were applied.

For homozygosity mapping, the genotype data from all five affected dogs were used. Markers with missing genotypes in one of the five cases and markers on the sex chromosomes were excluded. The output intervals were intersected with the intervals from the linkage analysis in an Excel spreadsheet to find overlapping regions (Appendix A). All positions correspond to the UU_Cfam_GSD_1.0 reference genome assembly, annotation release no. 106.

### 2.8. Whole Genome Sequencing

Illumina TruSeq PCR-free DNA libraries with ~400 bp insert size were prepared from two affected dogs (Dogs #1 and #5). A total of 217 million and 239 million 2 × 150 bp paired-end reads were collected (NovaSeq 6000 instrument, Illumina Inc., San Diego, CA, USA), corresponding to 22.5× and 24.7× coverage, respectively, and mapping and alignment were performed [24]. The sequence data were deposited under the study accession PRJEB16012 and the sample accessions SAMEA110175956 and SAMEA110175523 in the European Nucleotide Archive.

### 2.9. Variant Calling

Variant calling was performed using GATK HaplotypeCaller [25] in the gVCF mode [24]. To predict the functional effects of the called variants, the SnpEff [26] software, together with NCBI annotation release 106 for the UU_Cfam_GSD_1.0 genome reference assembly, was applied. For variant filtering, 784 control genomes from dogs of diverse breeds were used (Appendix A) [24]. Variants were filtered as private for the affected cases if the genotypes of the control dogs were either of the homozygous wild type or missing. No other ESSPs were among the control genomes for variant filtering.

### 2.10. Gene Analysis, PCR, and Sanger Sequencing

The numbering within the canine *EHBP1L1* gene corresponds to the NCBI RefSeq accession numbers XM_038563927.1 (mRNA) and XP_038419855.1 (protein). The primers 5′-GAT CTG GCA CAG GTA CGT CA-3′ (Primer F) and 5′-TTT TCC AAA ACC ACC CTT TG-3′ (Primer R) were used for the generation of an amplicon containing the *EHBP1L1*:c.3120delC variant. A smaller PCR product of a size of 130 bp (or 129 in the case of the mutant allele) was amplified from FFPE-derived DNA with Primer F and 5′-ATC AGC GCT CGG TCC CTA-3′ (Primer FFPE-R). PCR products were amplified from genomic DNA using AmpliTaq Gold 360 Master Mix (Thermo Fisher Scientific, Reinach, Switzerland). Direct Sanger sequencing of the PCR amplicons on an ABI 3730 DNA Analyzer (Thermo Fisher Scientific, Reinach, Switzerland) was performed after treatment with exonuclease I and alkaline phosphatase (BioConcept Ltd., Allschwil, Switzerland). The Sanger sequences were analyzed using the Sequencher 5.1 software (Gene Codes, Ann Arbor, MI, USA).

## 3. Results

### 3.1. Animals, Pedigree, and Family History

We examined 26 show ESSPs from a large family in Sweden. Five female ESSPs (Dogs #1–5) from two breeders and three different litters were considered to be clinically affected with DAMS based upon the breeders’ and owners’ assessments, physical examinations, and clinicopathological test results (Figure 1). An additional two puppies (Dogs #6 and 7) in a litter of nine died within the first day. Frozen tissue, but no further clinicopathological information, was available for these two puppies. According to the breeder of the other two litters, Dogs #1–3 were the only clinically affected puppies in their litters, but this breeder also recalled having neonatal deaths in two other litters from the same two parents (no further details and samples available).

According to the breeders, all dogs affected with DAMS appeared slightly stunted, less well muscled, and less active compared to their littermates during the first few weeks of life. No overt physical abnormalities were noted in the pre-sale veterinary examination at two months of age. However, the new owners of the puppies recognized exercise intolerance, weakness, and regurgitation from puppyhood. In addition, difficulties opening and closing the mouth, dysphagia, and difficulty in lapping water (Dogs #4 and 5), as well as aspiration pneumonia (Dog #1), seizures (Dog #5), and chronic diarrhea (Dog #1), were noted in the affected juvenile dogs. The affected dogs were 0.7 to 4 years of age at the initial examination and were followed over a period from two months to three years. The five affected dogs (Dogs #4, 5, 2, 3, and 1) were euthanized at approximately 1, 1.5, 5, 7, and 7.5 years of age, respectively, due to failure to thrive, progressive clinical signs, and poor quality of life.

The 19 examined dogs related to the affected dogs (Figure 1) were clinically healthy according to the owners’ reports and physical examinations. They did not show any muscular atrophy at the time of examination—between 1 to 10 years of age—or during the one-year follow-up period. The four show ESSPs that were unrelated within greater than six generations to the above family and presented at the clinic for other reasons had no abnormalities consistent with DAMS and were included as breed controls for hematological analysis and biochemistry only.

The pedigree analysis showed that the affected dogs, including the previously reported affected dogs from Australia [17], were all closely related with a common ancestor (eight generations for Dogs #1–3, ten generations for Dogs #4–7, and four generations for Dogs #A1–A3 from Australia; Figure 1). The common ancestor for all affected ESSPs from Sweden and Australia was born in ~1970 in England. This championed sire appears in many pedigrees of championed breeding ESSPs from England, Sweden, and Australia. However, there is no direct evidence that he carried or passed on the mutant allele, as no samples were available.

The pedigree analysis showed that both male and female dogs can present with DAMS, that the parents of affected dogs lacked clinical signs of the syndrome, and that all dogs affected with DAMS were related to a common ancestor. This is consistent with a monogenic autosomal recessive mode of inheritance.

### 3.2. Clinical Examination and Imaging Findings

In physical examinations, the five affected (Dog #1–5) dogs with DAMS from Sweden appeared slightly stunted and had severe muscle atrophy (muscle score 2–4/4 [4 normal], depending on localization), with the temporal and limb muscles being most prominently affected (Figure 2). All dogs had a body condition score of 4–6/9 [5 normal]. All affected dogs had clinical evidence (*n* = 5) and four had radiographical evidence of megaesophagus (Dogs #1–3 and 5). The dog without a radiographic diagnosis of megaesophagus (Dog #4) was necropsied within days of clinical examination, and megaesophagus was confirmed. Two affected dogs had trismus. None of the other related dogs had any clinical signs of a myopathy or anemia. The two affected ESSPs from Australia included in this study had similar clinical signs as those with DAMS from Sweden [17].

In addition, two male siblings (Dogs #6 and 7) of Dogs #4 and 5 (Figure 1) died during their first day of life without any overt malformations or other specific signs according to the breeder, but they were not further examined.

### 3.3. Hematological Abnormalities

All affected ESSP dogs (Dogs #1–5) had similar hematological abnormalities. The main CBC findings were a mild non-regenerative, microcytic hypochromic anemia (Table 1, Figure 3a–c and Figure 4a). Blood smear examination confirmed these findings with hypochromic erythrocytes, many target cells, acanthocytes, and marked metarubricytosis (4–297 nucleated red blood cells [RBCs]/100 leukocytes) with abnormal morphology (Figure 4a). The metarubricytes were microcytic, often with pyknotic nuclei that had small projections. A few binucleated metarubricytes were seen. Metarubricytes had relatively scant, lightly blue cytoplasm that occasionally contained small, round, blue structures resembling granuli (Figure 4a). Mature erythrocytes did not have inclusions (Pappenheimer bodies) or other morphological abnormalities, resembling those seen in metarubricytes. The corrected white blood cell (WBC) count and differential WBC picture, as well as the platelet number and parameters determined by the hematology analyzer, were within the normal reference intervals. The microscopic evaluation concurred with the counts and revealed no morphological abnormalities in platelets or myeloid and lymphoid cells. Similar hematological features were found in the two affected ESSPs that were included from Australia [17].

### 3.4. Serum Chemistry

The serum from the five dogs from Sweden with DAMS had higher phosphate and bilirubin concentrations compared to the related healthy ESSPs, but the values were within the published reference intervals for dogs (Table 2). Dog #2 had increased serum creatinine and urea concentrations shortly before being euthanized. All other serum chemistry results were comparable between affected and other ESSPs and were well within the normal reference intervals (Table 2).

### 3.5. Other Serum Tests

Additional tests performed in 2–4 affected dogs revealed negative acetylcholine receptor and antinuclear antibody titers, and baseline cortisol, total thyroxine (T4), thyroid stimulating hormone (TSH), C-reactive protein, lead, cobalamin, and folic acid concentrations were within the reference intervals. While the serum iron, unsaturated iron binding capacity, total iron binding capacity, iron saturation, and ferritin values were comparable to those of related ESSPs, the hepcidin and sTFR concentrations and, thus, sTFR/ferritin index, were abnormal in two of the three tested affected dogs (Table 3).

### 3.6. Gross Pathology and Histopathology

Two dogs (#2 and 4) with DAMS were necropsied. Dog #2 was an intact 5-year-old female show ESSP with fat deposition above normal and generalized moderate to severe muscle atrophy. Dog #4 was an intact one-year-old female show ESSP and had fat deposition within normal variation, but generalized moderate muscle atrophy. The temporal and masticatory musculature were most severely affected in both dogs. Dog #2 also had severely affected appendicular muscles. Both had a flaccid and dilated esophagus; that of Dog #2 was up to 4 cm in diameter and that of Dog #4 was up to 8 cm. Dog #2 also had an enlarged, rounded heart with a thickened septum and left ventricular wall, as well as dilated right atrium, pulmonary ostium, and aortal ostium. Dog #4 had no apparent heart lesions.

Striated muscle samples from *m. triceps brachii* (Dogs #1, 2, and 4), *m. biceps femoris* (Dogs #1, 2, and 4), and *m. masseter* (Dog #2 and 4) were examined histopathologically in more detail and with morphometry. Within all sections of each muscle from every dog, the myofibers appeared disorganized to various degrees, and they were found to be often shortened, and with marked variation in diameter and orientation (Table 4, Figure 5).

Dog #1, which had the mildest clinical signs, had only myocyte lesions without lipocyte infiltrates or fibrosis in the examined musculature (Table 4). In addition, the *m. biceps femoris* of Dog #1 had the highest proportion of myocytes with the central pallor, i.e., variably sized central areas of clear to pale eosinophilia and a finely granular to feathery cytoplasm. The central pallor was Periodic Acid Schiff negative. Dog #4 had the highest proportions of myocytes with one or more centralized nuclei (Table 4), and Dog #2 had a higher median myocyte diameter and the largest variation in myocyte size, as well as the most extensive lipocyte infiltrates. Degenerated myocytes that were swollen with vacuolated sarcoplasm and loss of cross-striations were seen in muscle samples from Dogs #2 and 4, and they were especially prominent in the *m. biceps femoris* of Dog #2, the oldest of all affected dogs examined. Occasional swollen hypereosinophilic myofibers were seen in all samples, and in a few samples, there were occasionally small foci of regenerative myocytes characterized by small amounts of basophilic cytoplasm, numerous linearly arranged internalized nuclei, prominent satellite cell nuclei at the cell margins. Variable adipocyte infiltrates—which were most severe in Dog #2—as well as endomysial and perimysial fibrosis were also present (Table 4, Figure 5). Notably, the myofiber lesions in Dog #1 were more synchronized and extensive than in Dogs #2 and 4 (Figure 5). There was minimal inflammation and necrosis in the examined skeletal muscle samples.

In dogs, the esophagus has a striated musculature along its entire length, with a smooth musculature admixed only in the part caudal to the diaphragm [27]. The histopathological examination of the esophagus (Dogs #2 and 4) showed the loss of myofibers in the muscularis mucosae and variable thinning of the inner and outer layer of the muscularis externa. The myofiber lesions were similar to those in the skeletal musculature, i.e., centralized nuclei and presence of regenerative and hypertrophic and fewer atrophic myocytes. There were minimal lipocyte infiltrates and minimal to mild fibrosis. There were no lesions in the epithelium or mucous glands.

The histopathological examination of the heart was unremarkable, as were the smooth muscles examined. No other organ and tissue abnormalities were noted in the gross and histopathological examinations, except for mild splenomegaly and a hyperplastic bone marrow, as described below.

### 3.7. Bone Marrow Evaluation

Bone marrow aspirates and core biopsy specimens were available from two affected dogs (Dogs #2 and 4, at 5 and 1 years of age, respectively). The bone marrow cytology showed normal to increased marrow cellularity (85–95%), with a normal to decreased myeloid to erythroid ratio (~1.0 and 0.03, respectively; normal ratio: 1.0–2.0 [28]). Dog #4 had a moderate number of partially lysed cells, which had the morphological features of erythroid cells. Erythroid maturation appeared complete, but an abundance of microcytic metarubricytes (~50% of the erythroid cells) with dysplastic changes in the form of pyknotic nuclei and a scant amount of cytoplasm was noted, in addition to approximately 5–10% bi- and multinucleated metarubricytes (Figure 4b,c). Around 93% of the marrow cells in Dog #4 were erythroid, and only 3% were myeloid.

Focal areas of macrophages with hemosiderin accumulation were found in both dogs. In contrast, the myelo- and megakaryopoiesis appeared normal with respect to the expected number of cells, maturation, proportion of precursor cells, and morphology. Similar results were reported in the three affected ESSPs from Australia [17].

An increased cellularity was seen with normal to high numbers of erythroid cells (50–80%), depending on the bone marrow region that was histopathologically examined. In the erythroid cell lineage, an appropriate complete maturation was present, but most erythroid cells were microcytic metarubricytes with morphologic abnormalities, as described above. Only 10–20% were myeloid cells, including a few polymorphonuclear cells. Both the megakaryocytic and myeloid cell lineages had normal morphologies and maturation. The number of mature megakaryocytes was normal to slightly increased. The occasionally interspersed fat cells were considered normal. There was no evidence of myelofibrosis. Around 8–10% of the myeloid cells were large mononuclear cells that were characterized as macrophages which were loaded with hemosiderin, and they often formed clusters. The abundant presence of tissue iron was confirmed by Perl’s staining (Figure 4d).

### 3.8. Genetic Analysis

The parametric linkage analysis in a litter with six living offspring and their parents (Appendix A) identified 43 linked segments spanning 277 Mb with a maximum LOD score of 1.10. The homozygosity mapping in the five affected dogs with DAMS identified 14 extended homozygous regions with shared haplotypes. A total of five genomic segments on five different chromosomes showed simultaneous linkage and homozygosity. Taken together, these five intervals spanned 9.0 Mb or roughly 0.38% of the 2.4 Gb dog genome, and they were considered the critical intervals for the subsequent analyses (Appendix A).

The entire genomes of two affected dogs (Dogs #1 and 5) were sequenced and searched for variants that were homozygous and shared by both cases, as well as those that were private in comparison to 782 control canine genomes (Table 5 and Appendix A).

The analysis of the entire genome identified a single homozygous private protein changing variant that was shared by both affected dogs. This variant was a single-base coding deletion in an exon of the *EHBP1L1* gene. It can be designated as Chr18:52,123,541delG (UU_Cfam_GSD_1.0 assembly) or XM_038563927.1:c.3120delC. The deletion is predicted to result in a frameshift and premature stop codon, thereby truncating 413 codons or 28% of the open reading frame encoding the wild-type EHBP1L1 protein (XP_038419855.1:p.(Phe1041Serfs*30)). No other private protein changing variants were identified in the genomes of the two affected dogs.

The presence of the deletion variant in *EHBP1L1* was confirmed through Sanger sequencing and genotyping of the five affected dogs with DAMS and the two dead puppies in the family from Sweden, as well as 13 unrelated and unaffected ESSPs (Figure 1 and Figure 6).

All five ESSPs with DAMS and the two ESSP puppies from Sweden with neonatal death, as well as the two tested ESSPs with DAMS from Australia [17], were homozygous for the mutant *EHBP1L1* allele. None of the 19 additionally examined healthy dogs in the family from Sweden were homozygous for this allele. The genotypes at the deletion co-segregated with the phenotype, as expected for a monogenic autosomal recessive mode of inheritance (Figure 1, Table 6). The three available parents, the available grandparent, and three healthy siblings of the affected dogs carried the mutant *EHBP1L1* allele in a heterozygous state.

## 4. Discussion

Hereditary diseases involving either congenital dyserythropoietic anemias [9,29,30] or congenital myopathies [10,11,31,32,33] have been described and associated with specific gene defects in humans and animals. In this study, we further characterize a dyserythropoietic anemia and myopathy syndrome (DAMS) in a large family of ESSP dogs as an autosomal recessive trait that is likely caused by an *EHBP1L1* frameshift deletion. Furthermore, two puppies that died neonatally also carried the mutant *EHBP1L1* allele in a homozygous state. Previously, EHBP1L1 deficiency was only associated with neonatal death in one murine knockout model [34] and with fetal death in two human pregnancies [35].

*EHBP1L1* codes for the Eps15-homology (EH) domain-binding protein 1-like 1. The EH domain is a module of ~100 amino acids originally identified in the tyrosine kinase substrate Eps15 that can mediate diverse protein–protein interactions [36]. The EH domain proteins are involved in regulating endocytosis and vesicle transport. Indeed, the EHBP1L1 protein links RAB8 and the BIN1–dynamin complex to generate membrane curvature and excises the vesicle at the endocytic recycling compartment for apical transport [34]. The protein is ubiquitously expressed, including in muscle and hematopoietic cells, but its function in muscle and erythropoietic cells has not been investigated. Overall, there is surprisingly little known about the EHBP1L1 protein in terms of health and disease in humans and animals, which is likely because of the lack of survival of animal models for further investigating its functions.

The one-base deletion in *EHBP1L1* found in the ESSPs with DAMS or neonatal death causes a frameshift and, thereby, a premature stop codon. While a severely truncated protein (lacking 413 amino acids or 28% of the open reading frame) is predicted, it is more likely that such an early stop codon will result in premature mRNA decay and, thus, no protein production [37]. However, no protein and expression studies were pursued here.

From a disease perspective, very little is known about EHBP1L1 deficiency. Homozygous *EHBP1L1^−/−^* knockout mice died neonatally and exhibited anemia and shortened small intestinal microvilli at birth [34]. In humans, homozygosity and compound homozygosity for truncating *EHBP1L1* variants were reported as the cause for non-immune hydrops fetalis resulting in fetal loss in two separate consanguineous families from Saudi Arabia [35]. In our study of an ESSP dog family, the two puppies (in a litter of nine) that died shortly after birth were homozygous for the *EHBP1L1* variant. Although neonatal deaths were observed by the other breeder of the affected litters in Sweden, no details or additional samples were available to study. These neonatal losses indicate that *EHBP1L1* deficiency could also be a cause of neonatal death in dogs, but the specific cause of death was not determined. It is highly unusual that a single gene variant is clinically associated with either perinatal deaths or a slowly progressive myopathy and dyserythropoiesis. Further studies are warranted to clarify whether *EHBP1L1* deficiency alone is sufficient to cause neonatal losses in some dogs and what might lead to the striking phenotypic differences with respect to DAMS.

It should be noted that there are many acquired and hereditary causes of perinatal deaths in humans and animals. In a large survey from Norway, 2.8% of newborn ESSP puppies died, while the overall early neonatal mortality for all canine breeds studied was 3.7% [38]. Unfortunately, only rarely do these losses undergo further clinical examination or have their underlying cause, such as congenital malformations [39] or maternal and environmental effects [38], identified. Nevertheless, the perinatal deaths associated with *EHBP1L1* gene defects observed in two human fetuses, knockout mice, and now, in this report of two neonatal puppy mortalities support causality.

In addition to the potentially *EHBP1L1*-related neonatal deaths, EHBP1L1 deficiency is also likely responsible for a unique dyserythropoietic anemia and progressive myopathy syndrome in ESSP dogs, which we term DAMS.

The observed non-regenerative anemia with metarubricytosis, target cells, and acanthocytes, as well as the marked erythroid hyperplasia and dyserythropoiesis with non-orderly maturation of erythrocytes and inappropriate metarubricytosis in ESSPs with DAMS, are morphologically consistent with what can be seen with congenital dyserythropoietic anemias (CDAs) in humans [29,30]. Based upon erythroid morphologies, genes involved, and modes of inheritance, there are several types of CDAs in human patients, but as yet, none have been associated with EHBP1L1 deficiency or its related pathways [29,30]. Moreover, a few CDAs are syndromic, but only rarely do they involve skeletal muscle in humans [30]. For instance, congenital megaloblastic macrocytic anemia and mitochondrial dysfunction were reported with *SFXN4*-related myopathy [40]. Severe sideroblastic anemia, including Pearson marrow–pancreas syndrome, can accompany congenital to childhood onset mitochondrial myopathy [41]. The *EHBP1L1* knockout mice dying perinatally were anemic, but were not further evaluated hematologically [30]. Likewise, the two human fetal deaths related to *EHBP1L1* deficiency were not examined for dyserythropoeisis or myopathy [35]. Some human CDAs can also be hemolytic [29], which, based upon the increased serum bilirubin concentrations, in comparison to that of related ESSPs, may have been present at a low grade in the ESSPs with DAMS. However, there was no reticulocytosis, and no erythrocyte survival studies were performed.

The four major types of CDA are macrocytic [37], while the dyserythropoietic anemia in the affected ESSPs with DAMS was microcytic and hypochromic with microcytic metarubricytes. Microcytosis and hypochromia are most commonly associated with iron deficiency. However, the affected ESSPs in this and the prior report [17] showed more than adequate amounts of tissue iron. In light of the adequate iron stores, the normal routine serum iron parameters and the abnormal serum transferrin receptor and hepcidin values in the affected ESSPs studied here may suggest membrane effects with the premature degradation of erythroid precursor and erythroid cells. One might speculate that EHBP1L1 deficiency affects the iron metabolism and cellular stability of erythroid cells, but the precise mechanism of the microcytic anemia in dogs with DAMS remains unknown. It is noteworthy that human CDA patients with Majeed syndrome caused by *LPIN2* variants have a hypochromic microcytic anemia with dyserythropoeisis, but also chronic recurrent multifocal osteomyelitis and inflammatory dermatosis [42]. In addition, siderocytic anemias in humans are microcytic [43]. Human patients with CDA frequently have splenomegaly [33], which was noted in some but not all affected ESSPs and in one of two Labrador retriever dogs with a similar syndrome [18].

The clinical and muscle studies in the affected ESSPs from Australia [17] and Sweden that were reported here, indicate an early onset and slowly progressive atrophic myopathy. Vague clinical muscle signs were seen in puppyhood, and thereafter, all affected dogs developed progressive temporal and generalized muscle atrophy, leading to humane euthanasia as young adults up until middle age. Muscle biopsies revealed centralized nuclei, central pallor, lipocyte infiltrates, and fibrosis. Thus, the myopathy could be considered a centronuclear myopathy (CNM), which is grouped under congenital myopathies [31,32]. However, DAMS is syndromic and is likely caused by a novel gene defect in dogs. Congenital myopathies can be divided depending on the presence of cores (core myopathy), central nuclei (centronuclear myopathy), or nemaline bodies (nemaline myopathy) [31,32]. Much progress has been made among the CNMs through the identification of specific gene defects involving *BIN1*, *CCDC78*, *DNM2*, *MYF6*, *MYMR14*, *RYR1*, *TTN*, *SPEG*, and *ZAK* and their pathways [31,32]. While an *EHBP1L1* gene defect has not been previously associated with CNM, it is not unexpected. As mentioned above, EHBP1L1 is an integral part of RAB8 and the BIN1–dynamin complex and could thus readily explain the myopathy [44,45,46,47]. Defects in BIN1 and dynamin 2 have been reported to cause CNM in humans and animals [12,13,14,31,32,33,47].

EHBP1L1 is also involved in regulating GLUT4 translocation and glucose uptake in myocytes through its interactions with TBC1D1 [48]. Given that neonatal puppies are very sensitive to hypoglycemia, it is possible that *EHBP1L1^−/−^* neonates are at increased risk for functional hypoglycemia that is secondary to the dysfunctional regulation of GLUT4 translocation and glucose uptake in the brain, musculature, and fat [49,50]. Furthermore, EHBP1L1 regulates the formation of tubular recycling endosomes in HeLa cells via interactions with GTP-bound Rab10 [51]. These are the same pathways that dynamin 2 affects via its control of actin polymerization [52]. Additionally, a CNM mouse model expressing mutant dynamin 2 actin filament disorganization impaired GLUT4 translocation to the plasma membrane, and in affected muscle biopsies, there was abnormal perinuclear GLUT4 accumulation [44]. In adult DNM2 knockout mice, the CNM mutant dynamin 2 localized around centralized nuclei, and the nuclear size and number were reduced [45]. In addition, dynamin 2 is one of the proteins that regulates nuclear peripheral positioning and triad organization during actin polymerization and myofiber T-tubule biogenesis [53,54]. Dogs with DAMS may offer a faithful model for investigating the mechanisms and pathophysiology of EHBP1L1 deficiency.

The observed megaesophagus causing dysphagia and regurgitation in affected ESSPs with DAMS can be explained by the fact that the upper part of the canine esophagus has striated muscle [27] and, thereby, is also affected by this myopathy. Indeed, the histopathological muscle features of the megaesophagus were like those of appendicular muscles and were consistent with CNM.

Interestingly, some affected ESSPs also had evidence of dilated cardiomyopathy, suggesting that cardiac muscle may also be affected by EHBP1L1 deficiency. However, no specific morphological changes suggestive of centronuclear myopathy were found in the hearts of the two necropsied dogs or those from Australia [17]. In fact, BIN1- and DNM2-related myopathies affect the skeletal muscular triad, while the heart has dyads and would thus be less affected. Since the clinical signs of skeletal myopathy progressed slowly, dilated cardiomyopathy may be a more subtle and slowly progressing finding or may be associated with other or secondary effects of EHBP1L1 deficiency.

Initially, it might sound surprising that a presumptive hereditary disease syndrome described in three ESSPs from Australia in 1991 was not seen or reported for three decades until this study on show ESSPs from Sweden with DAMS or neonatal deaths that were homozygous for the *EHBP1L1* variant. However, it is not uncommon that recessive disease alleles segregate in a breed and go unnoticed for several decades [55,56], which underscores the need for the introduction of genetic testing and targeted breeding programs to combat hereditary diseases in purebred animals. Further, breeders infrequently seek veterinary medical attention when faced with neonatal deaths and unthrifty puppies (known as puppy morbidity and mortality complex), and they accept these losses. It is also possible that other affected dogs were seen but not recognized and/or not reported. Moreover, as occurs often in dog breeding with international travel, the show ESSPs in Australia were related to the ones in Sweden within only a few generations. The available pedigree information allowed the tracing of both the paternal and maternal sides of all litters to a distant common ancestor, thereby supporting an autosomal recessive trait. The late common ancestor was a champion with many offspring, but no samples were available to determine if he was indeed a carrier for the *EHBP1L1* variant. There is no overt advantage of show ESSPs being carriers for the *EHBP1L1* variant and, thus, no evidence of positive selection for the mutant variant.

It is currently unknown how widespread the mutant *EHBP1L1* allele among show ESSPs in Sweden, Australia, and the world is. Maybe some to most homozygous affected animals succumb to perinatal death or fail to thrive as young puppies and, thus, do not receive veterinary attention and further clinicopathological and genetic evaluations. Future genotyping set up by commercial genetic testing laboratories will allow for simple screening of show ESSPs with suspicious signs or prior to breeding and, thereby, allow for a definitive diagnosis and prevention of the breeding of any affected animals or of carriers with carriers, avoiding the production of any affected animals in future generations. Additional investigations in dogs with this naturally occurring syndrome may help further elucidate the various EHBP1L1 protein functions and pathophysiology of this syndrome.

Utilizing the ACMG/AMP criteria applied in human medical genetics [57], the loss-of-function variant identified in the ESSPs with DAMS or neonatal death constitutes very strong evidence for pathogenicity (PVS1). We rated the perfect co-segregation of genotypes in a very large pedigree and the absence of the mutant allele in nearly 800 control dogs as moderate evidence for pathogenicity (PM2). Taken together, we believe that the evidence from the current study is sufficient to classify this variant as very likely pathogenic according to the ACMG/AMP criteria. During this study, we became aware of an independent effort to investigate the genetic defect in Labrador retrievers with a similar syndrome [18]. Shelton et al. reported the identification of an *EHBP1L1* nonsense variant as a likely cause for dyserythropoietic anemia and polymyopathy in this breed [58]. The fact that two independent truncating variants in the same gene lead to virtually identical and highly characteristic clinical phenotypes in ESSPs and Labrador retrievers provides a further proof for the causality of these variants.

## 5. Conclusions

We identified an *EHBP1L1* frameshift deletion as a cause of DAMS and, possibly, neonatal loss in a large family of ESSP dogs. While EHBP1L1 deficiency is perinatally lethal in humans and knockout mice, dogs with DAMS can present as neotal deaths but also represent a different and unusual syndromic phenotype with much longer survival. Our findings considerably expand the known genotype–phenotype correlations for this yet not well studied gene. Additional investigations in dogs with this naturally occurring disease model may help to further elucidate *EHBP1L1* function in different tissues and the pathophysiology of this syndrome. Furthermore, our findings enable the genetic testing of ESSPs for early diagnosis and disease prevention through targeted breeding strategies.

## Figures and Tables

**Figure 1 genes-13-01533-f001:**
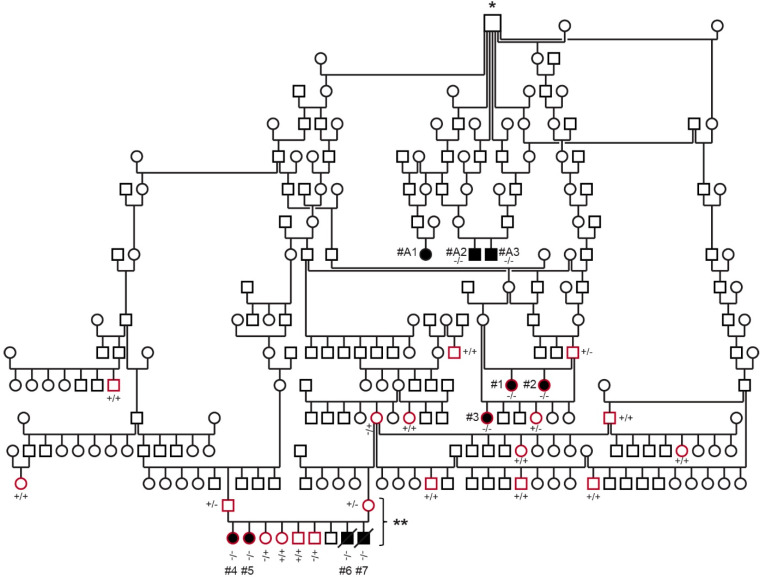
Pedigree and genotyping of the ESSP family with DAMS. Squares are males, and circles are females; affected dogs are indicated with black-filled symbols, and open symbols represent unaffected dogs (or dogs without phenotype information); a diagonal line represents perinatal death. The dogs circled in red were clinically examined and genotyped during this study. Dogs #A1–A3 represent the DAMS-affected dogs from Australia, which were previously reported in [17]. Genotypes for the *EHBP1L1*:c.3120delC with mutant (-) and wild type (+) allele are indicated for dogs from which DNA samples were available. The symbol marked by * represents the first known common ancestor of all investigated affected dogs. The litter marked with ** comprising both parents and six available offspring (excluding dead puppies) was used for parametric linkage analysis.

**Figure 2 genes-13-01533-f002:**
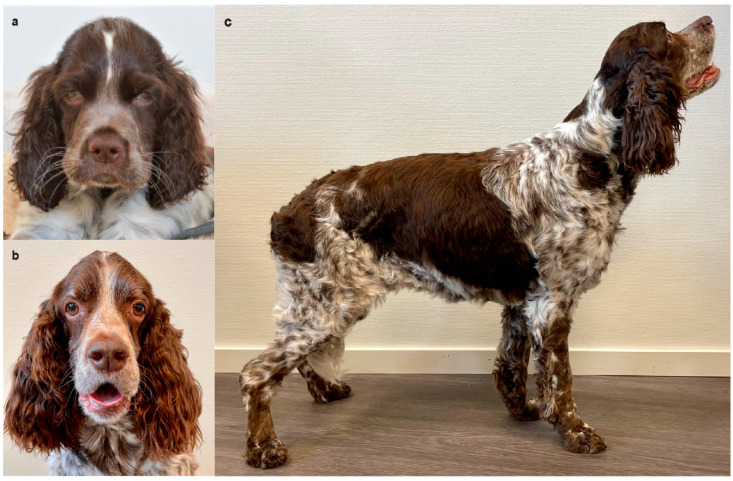
Affected English springer spaniel Dog #1 (**a**) as a puppy and (**b**,**c**) at seven years of age, showing severe temporal and pelvic limb muscle atrophy. Additionally noted are the wider positioning of the eyes, drooping lips, lower placement of the ears, and tilted stance.

**Figure 3 genes-13-01533-f003:**
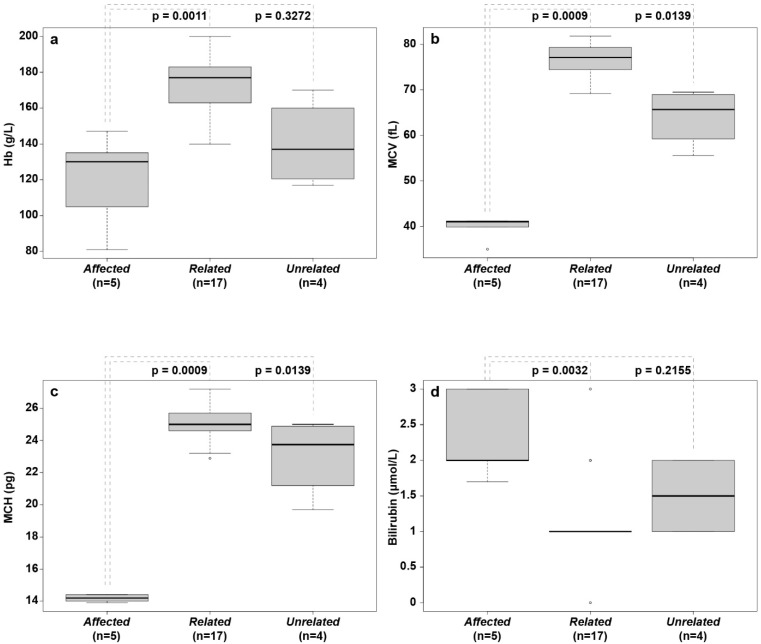
Box plots of hematological parameters and serum bilirubin concentration from affected, related, and unrelated English springer spaniels. (**a**) Hemoglobin concentration (Hb/g/L); (**b**) mean corpuscular volume (MCV/fL); (**c**) mean cell hemoglobin (MCH/pg; (**d**) bilirubin concentration (µmol/L). The median values are indicated by horizontal lines in the boxes, where the first to third quartiles define the boundaries. Whiskers extend to the minimum and maximum values. The affected dogs had significantly lower MCV and MCH values than those of both other groups, and they had lower Hb than that of related healthy dogs.

**Figure 4 genes-13-01533-f004:**
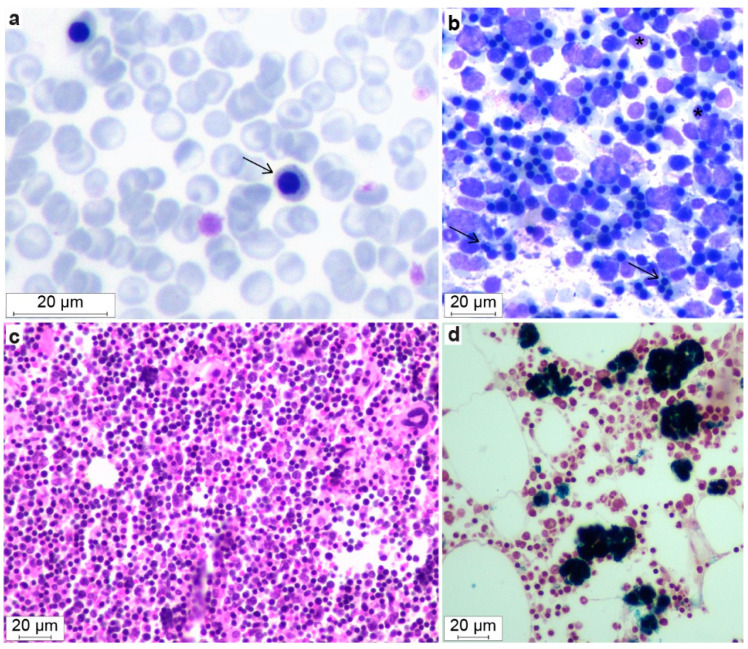
Morphological features of blood smears and bone marrow in dogs with DAMS. A blood smear (**a**) with microcytic metarubricytes (arrow) showing abnormal morphology with blue granula-like structures in the cytoplasm (May–Grünwald–Giemsa). Marrow cytology (**b**) with an increased number of erythroid cells. Note the microcytic metarubricytes with small amounts of cytoplasm, binucleated metarubricytes (black arrow), and pyknotic nuclei (*) (May–Grünwald–Giemsa). Histopathology of bone marrow (**c**) that also shows the hypercellularity and erythroid hyperplasia (Meyer’s H&E) and (**d**) focal areas of macrophages with hemosiderin accumulations (Perl’s stain positive).

**Figure 5 genes-13-01533-f005:**
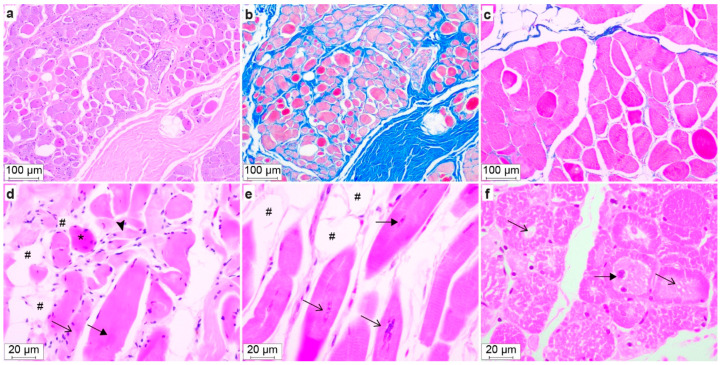
Histopathology of muscle sections from Dogs #2 (**a**–**d**), #4 (**e**), and #1 (**f**). H&E staining (**a**,**d**–**f**) and Masson trichrome staining (**b**,**c**). (**a**,**b**) *m. masseter* with severe endomysial and perimysial fibrosis seen as blue staining in (**b**). (**c**) *m. biceps femoris* with sparse blue-stained areas, i.e., no fibrosis. (**d**) *m. triceps brachii* showing disorganized myofibers, hypertrophic (filled arrow), normotrophic (arrow), atrophic (arrowhead), hypereosinophilic swollen myofibers (*), and lipocyte infiltrates (#). (**e**) *m. triceps brachii*, longitudinal section, showing centralized nuclei (filled arrow), chains of nuclei (arrow) and lipocyte infiltrates (#). (**f**) *m. biceps femoris*, transverse section, showing variable myocyte diameter, centralized nuclei (filled arrow) and central cytoplasmic pallor (arrow).

**Figure 6 genes-13-01533-f006:**
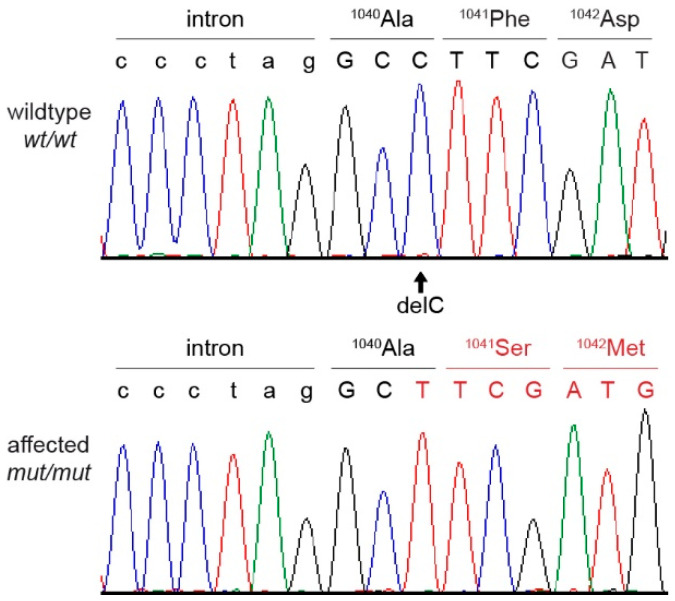
Sanger sequencing chromatograms surrounding the *EHBP1L1*:c.3120delC variant and the amino acid sequence of a wild type and an English springer spaniel with DAMS. A homozygous deletion of a single cytosine is visible in the affected dog, resulting in a frameshift with an early stop codon.

**Table 1 genes-13-01533-t001:** Hematology results for affected, related, and unrelated ESSP dogs from Sweden with dyserythropoietic anemia and myopathy syndrome.

Analytes	Affected (*n* = 5) Median (Range)	Related (*n* = 17) Median (Range)	Unrelated (*n* = 4) Median (Range)	Reference Intervals ^a^
RBCs (10^12^/L)	9.16 (5.64–10.60)	6.84 (5.93–8.62)	6.28 (5.01–9.80)	6.2–8.9
HCT (%)	37.3 (23.1–38.0) ^b^	53.4 (45.0–62.1)	38.0 (33.0–47.2)	41–57
Hb (g/L)	130 (81–147) ^b^	177 (140–200)	137 (117–170)	145–200
MCV (fL)	41.0 (35.0–41.2) ^b^	77.10 (69.2–81.8)	65.65 (55.6–69.5)	57–73
MCH (pg)	14.2 (13.9–14.4) ^b^	25.0 (22.9–27.2)	23.8 (19.7–25.0)	21–25
MCHC (g/L)	351 (339–391)	332 (297–349)	361 (355–362)	335–380
RET (10^9^/L)	68 (45–150)	56 (32–139)	60 (13–130)	60–150
PLT (10^9^/L)	248 (148–380)	199 (105–321)	273 (164–505)	108–562
nRBC/100 leukocytes	79 (4–297) ^b^	0 (0–0)	0 (0–0)	<1
WBC (10^9^/L)	7.8 (4.4–17.0)	9.5 (5.7–20.0)	6.9 (3.6–21.3)	4.9–14.7
Seg. neutrophils (10^9^/L)	5.9 (3.5–9.8)	6.0 (4.1–16.9)	4.7 (2.1–11.0)	2.3–11.3
Band neutrophils (10^9^/L)	0.5 (0.1–1.0)	0.2 (0.0–2.0)	0.1 (0.0–1.3)	0–0.3
Lymphocytes (10^9^/L)	2.2 (0.3–4.8)	2.4 (0.6–4.8)	1.60 (0.6–3.6)	1.2–5.7
Monocytes (10^9^/L)	0.2 (0.10–0.6)	0.3 (0.1–0.6)	0.40 (0.0–1.7)	0.1–1.3
Eosinophils (10^9^/L)	0.2 (0.1–1.7)	0.3 (0.0–1.1)	0.3 (0.1–0.6)	0–1.7
Basophils (10^9^/L)	0 (0–0)	0 (0–0)	0 (0–0)	0–0.1

^a^ Reference interval from Sysmex xt2000i (Sysmex Nordic, Landskrona, Sweden). ^b^
*p*-value of ≤0.002 is significantly different compared to the related ESSPs; Kruskal–Wallis. In addition, the values are different from those of the control ESSPs and canine reference intervals. Red blood cells (RBCs), hematocrit (HCT), hemoglobin (Hb), mean corpuscular volume (MCV), mean cell hemoglobin (MCH), mean corpuscular hemoglobin concentration (MCHC), absolute reticulocyte count (RET), nucleated red blood cells (nRBCs), white blood cells (WBCs), segmented neutrophils (Seg. neutrophils).

**Table 2 genes-13-01533-t002:** Serum chemistry results from affected, related, and unrelated English springer spaniels.

Serum Parameters	Affected (*n* = 5) Median (Range)	Related (*n* = 17) Median (Range)	Unrelated (*n* = 4) Median (Range)	Reference Intervals ^a^
Bilirubin (μmol/L)	2.0 (1.7–3.0) ^b^	1.0 (0.0–3.0)	1.5 (1.0–2.0)	<2.8
Cholesterol (mmol/L)	6.8 (6.7–7.5)	5.6 (4.2–8.8)	5.8 (5.4–11.0)	3.0–10.3
Triglycerides (mmol/L)	1.0 (0.7–1.7)	1.6 (0.5–5.6)	1.1 (0.7–1.4)	0.5–2.7
Glucose (mmol/L)	4.9 (4.5–6.3)	3.8 (3.1–5.2)	4.2 (3.4–5.3)	3.7–6.6
Urea (mmol/L)	5.0 (4.0–33.0)	6.0 (4.0–10.0)	4.0 (3.0–5.0)	3.0–9.0
ALP (U/L)	1.2 (0.5–4.8)	0.8 (0.3–2.1)	0.8 (0.4–4.1)	<1.4
ALT (U/L)	0.5 (0.3–0.8)	0.6 (0.3–1.2)	1.4 (0.6–11.7)	<1.2
AST (U/L)	0.5 (0.-1.20)	0.4 (0.2–2.0)	0.5 (0.4–17.1)	<1.0
Creatine kinase (U/L)	2.2 (0.9–2.6)	1.9 (1.0–38.0)	2.3 (1.0–8.5)	<3.7
Creatinine (μmol/L)	44 (35–246)	71 (42–125)	65 (43–81)	<135
Albumin (g/L)	38 (30–44)	34 (28–42)	29 (21–33.3)	30–45
Total protein (g/L)	63.0 (60.0–73.0)	46 (58.5–63.6)	53.0 (41.0–81.0)	49.0–71.0
Calcium (mmol/L)	2.8 (2.6–3.0)	2.6 (2.0–2.8) ^c^	2.5 (2.0–2.8)	2.4–3.0
Phosphate (mmol/L)	1.5 (1.3–2.5) ^b^	1.3 (0.9–1.6)	1.2 (0.8–1.4)	0.7–1.9
Sodium (mmol/L)	145 (137–150)	141 (127–150)	134 (121–145)	138–149
Potassium (mmol/L)	4.6 (3.9–5.1)	4.5 (4.0–5.1) ^c^	4.1 (3.8–4.8)	3.4–4.8
Chloride (mmol/L)	109 (101–111)	105 (94–114)	101 (91–110)	98–119
CRP (mg/L)	6.9 (5.3–10.6) ^d^	2.3 (1.3–65.6) ^d^	ND	<15

^a^ Reference interval from Cobas C311 (Roche Diagnostics Scandinavia, Bromma, Sweden). ^b^
*p*-value of ≤0.05 indicates a significant difference compared to related ESSPs; ^c^ serum calcium and potassium values were not available for one unrelated dog due to analytical error (*n* = 16). ^d^
*n* = 4. Alkaline phosphatase (ALP), alanine aminotransferase (ALT), aspartate aminotransferase (AST), C-reactive protein (CRP), not done (ND).

**Table 3 genes-13-01533-t003:** Serum iron parameters in English springer spaniels with DAMS and related healthy dogs.

Serum Parameters	Affected (*n* = 4) Median (Range)	Related (*n* = 4) Median (Range)	Reference Interval
Iron (µmol/L)	39 (19–54)	28 (18–43)	15–45
UIBC (µg/dL)	155 (127–258)	240 (182–327) ^a^	NA
TIBC (µg/dL)	402 (294–442)	423 (362–522) ^a^	260–430
Iron Saturation (%)	58 (30–70)	37 (34–57) ^a^	20–60
Ferritin (µg/L)	192 (170–212)	180 (129–217) ^a^	NA
sTFR (mg/L)	66.5 (22.6–116.2) ^b^	10.1 (9.5–12.8) ^a^	NA
sTFR–Ferritin Index	28.9 (9.7–51.8) ^b^	4.8 (4.1–5.69) ^a^	NA
Hepcidin (ng/mL)	9.1 (8.5–30.4) ^a,b^	37.7 (12.9–59.8)	NA

^a^*n* = 3, ^b^
*p*-value of <0.05 indicates significant difference, unsaturated iron binding capacity (UIBC), total iron binding capacity (TIBC), and soluble transferrin receptor (sTFR).

**Table 4 genes-13-01533-t004:** Myocyte diameter and evaluation of centralized nuclei, clarity, lipocyte infiltration, and fibrosis in three English springer spaniels with dyserythropoietic anemia myopathy syndrome.

Dog	Striated Muscle	Myocyte Diameter, µm ^a^	Centralized Nuclei ^b^	Central Pallor ^b^	Lipocyte Infiltrate	Fibrosis
#2	*m. triceps brachii*	29.0 (12.5–52.9)	19	3	severe	no
	*m. biceps femoris*	41.3 (13.0–84.8)	11	15	mild	no
	*m. masseter*	23.3 (10.1–47.4)	23	3	moderate	moderate
#4	*m. triceps brachii*	26.5 (14.7–47.1)	28	4	mild	no
	*m. biceps femoris*	28.2 (12.9–55.4)	25	7	no	no
	*m. masseter*	15.1 (7.2–42.9)	52	3	moderate	moderate
#1	*m. triceps brachii*	34.5 (15.2–58.9)	2	0	no	no
	*m. biceps femoris*	28.6 (10.2–56.7)	20	39	no	no

^a^ Median [Range]. ^b^ Percent (affected myocytes per 100 myocytes). Dogs #2 and 4 had more severe changes, which corresponded well with the more severe clinical signs observed.

**Table 5 genes-13-01533-t005:** The results of homozygous variant filtering in two affected English springer spaniels with DAMS against 784 control genome sequences.

Filtering Step	Homozygous Variants	
	In Whole Genome	In Critical Interval
Shared variants in two affected dogs	1,659,097	14,272
Private variants in two affected dogs	112	15
Private protein changing variants	1	1

**Table 6 genes-13-01533-t006:** Genotype–phenotype association in the *EHBP1L1*:c.3120delC variant with DAMS and neonatal death in 41 English springer spaniel dogs.

Dogs and Phenotype	*n*	*EHBP1L1* Genotype		
		+/+	+/del	del/del
Family from Sweden (*n* = 26)				
Dogs affected by DAMS	5	-	-	5
Puppies with neonatal death ^a^	2	-	-	2
Clinically unaffected relatives	19	12	7	-
Dogs affected with DAMS from Australia ^b^	2	-	-	2
Unrelated dogs from Switzerland/Germany ^a^	13	13	-	-

^a^ No clinicopathological examination was performed. ^b^ Clinical signs were previously reported [17].

## Data Availability

The accessions for the sequence data reported in this study are listed in Appendix A.

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
