# Peer review of "EHBP1L1 Frameshift Deletion in English Springer Spaniel Dogs with Dyserythropoietic Anemia and Myopathy Syndrome (DAMS) or Neonatal Losses"

_genes, 2022, doi:10.3390/genes13091533_

Round 1

Reviewer 1 Report

The current study "EHBP1L1 Frameshift Deletion in English Springer Spaniel  Dogs with Dyserythropoietic Anemia and Myopathy Syndrome (DAMS) or Neonatal Losses" is well-designed and presented.

Author Response

Reviewer 1 did not make any specific requests for changes to the manuscript. We thank the reviewer for the positive evaluation

Reviewer 2 Report

This study has emphasized a concept of “Genetic pleiotropism” that a mutation in one single gene could cause hereditary diseases in purebred dogs with a variety of clinical signs. The Sanger sequencing results of the EHBP1L1 gene of the affected dogs revealed only one deletion of the EHBP1L1 gene that causes frameshift mutation and premature stop codon. This EHBP1L1 frameshift deletion was proposed as a cause of dyserythropoietic anemia and myopathy syndrome (DAMS) and neonatal loss in the English springer spaniel dogs (ESSPs). A single gene mutation was confirmed in the affected dogs that this important discovery could increase the chances of having genetic abnormality tests in breeders to reduce number of neonatal loss and congenital diseases in dogs and it is useful for people studying hereditary diseases in dogs worldwide. To the best of my knowledge, the methods appear excellent.

Comments:

Materials and Methods

The Illumina sequencing platform was used for the genome sequencing process of this study. Two affected dogs were selected for the genome sequencings.

1) Does it affect genome analysis using only two genome sequences retrieved from two affected dogs?

2) If the genome sequencing is done in all affected dogs, Will it yield better results?

3) Could the authors explain further in choosing dogs number one and number five?

4) The NCBI reference sequences used in this study were removed as the results of standard genome annotation processing and the reference sequences were specified as obsolete version. Is there any effects to the results of this study?

Author Response

Thank you very much for the compliments!

Comments:

Materials and Methods

The Illumina sequencing platform was used for the genome sequencing process of this study. Two affected dogs were selected for the genome sequencings.

1) Does it affect genome analysis using only two genome sequences retrieved from two affected dogs?

Response: This study could have theoretically been based on the genome sequences of 1, 2, 3, 4 or all 5 of the available DAMS cases. Our rationale for choosing 2 affected dogs for whole genome sequencing was based on a compromise of cost considerations (fewer genomes are cheaper to sequence) and a sufficient power to exclude neutral variants that are not causally related to the trait (more sequenced genomes will increase the power). Retrospectively, our choice turned out to be ideal. The variant filtering using 2 case genomes and 782 control genomes yielded only a single private protein-changing variant that was shared by both case genomes and absent from the control genomes (Table 5). This number would have stayed the same, if we had sequenced 3, 4 or all 5 case genomes. Thus, in this specific project, the inclusion of additional case genomes would not have made a tangible difference to the outcome of the study. (Additional remark: Mostly due to cost considerations, the majority of previously published studies on monogenic diseases in dogs actually used only 1 case genome for comparable analyses.)

As our manuscript is focused on the biology of the trait and not on a systematic methodologic evaluation of using different numbers of cases (and/or controls), we did not change the manuscript with respect to this comment.

2) If the genome sequencing is done in all affected dogs, Will it yield better results?

Response: Similar to the comment above. If we had sequenced the genomes of all 5 affected dogs, the results and conclusions of our study would not have changed.

3) Could the authors explain further in choosing dogs number one and number five?

Response: We chose 2 cases that were as distantly related as possible to minimize the proportion of genome that is shared identical by descent (IBD) between relatives. We also had to consider the quality and quantity of the available DNA samples.

4) The NCBI reference sequences used in this study were removed as the results of standard genome annotation processing and the reference sequences were specified as obsolete version. Is there any effects to the results of this study?

Response: NCBI selects one individual genome assembly and its newest annotation for each species to define their preferred set of reference sequences (RefSeqs). However, in the last 2 years, several different high quality assemblies of different individual dogs were produced (e.g. a male Labrador Retriever, a female Boxer, a female German Shepherd etc.). In their latest annotation release 106 (from 8th of January 2021), NCBI performed independent annotations on 5 available individual dog genomes. NCBI selected the Labrador Retriever assembly (ROS_Cfam_1.0) as their preferred assembly. Consequently, all other transcript annotations derived from the other four genomes are now flagged by NCBI as "obsolete". This terminology is somewhat misleading as the transcript annotations on the other dog genomes were derived with exactly the same methodology, quality standards and at the same time as the Labrador Retriever annotation.

We consciously chose the German Shepherd assembly (UU_Cfam_GSD_1.0) as reference for our study as this assembly was chosen as reference by the global dog10K consortium (http://www.dog10kgenomes.org/). The differences between the EHBP1L1 transcript annotations on the ROS_Cfam_1.0 and the UU_Cfam_GSD_1.0 assemblies are minimal. The Labrador Retriever EHBP1L1 isoform 1 transcript contains 5065 nucleotides (XM_038424987.1) and the corresponding German Shepherd transcript contains 5066 nucleotides (XM_038563927.1). Both transcripts encode exactly the same 1472 aa protein (XP_038419855.1 vs XP_038280915.1). The minimal differences in the nucleotide sequences most likely represent neutral genetic variation between the two individual dogs that were sequenced to produce the genome assemblies.

The conclusions of our study (including the variant designation on the cDNA and protein level) would not have changed, if we had used the ROS_Cfam_1.0 genome assembly and its corresponding annotation instead of the UU_Cfam_GSD_1.0 genome assembly and its annotation.